# Free fermion entanglement with a semitransparent interface: the effect of graybody factors on entanglement islands

**Jorrit Kruthoff,**[1] **Raghu Mahajan,**[1] **and Chitraang Murdia.**[2,3]

[1] Department of Physics, Stanford University, Stanford, CA 94305-4060, USA

[2] Berkeley Center for Theoretical Physics, Department of Physics, University of California, Berkeley, CA 94720, USA

[3] Theoretical Physics Group, Lawrence Berkeley National Laboratory, Berkeley, CA 94720, USA

kruthoff@stanford.edu, raghumahajan@stanford.edu, murdia@berkeley.edu

## Abstract

We study the entanglement entropy of free fermions in 2d in the presence of a partially transmitting interface that splits Minkowski space into two half-spaces. We focus on the case of a single interval that straddles the defect, and compute its entanglement entropy in three limits: Perturbing away from the fully transmitting and fully reflecting cases, and perturbing in the amount of asymmetry of the interval about the defect.

Using these results within the setup of the Poincaré patch of AdS$_2$ statically coupled to a zero temperature flat space bath, we calculate the effect of a partially transmitting AdS$_2$ boundary on the location of the entanglement island region. The partially transmitting boundary is a toy model for black hole graybody factors. Our results indicate that the entanglement island region behaves in a monotonic fashion as a function of the transmission/reflection coefficient at the interface.

# 1 Introduction and summary

## 1.1 Entanglement entropy for 2d free fermions

Entanglement entropy is an important quantity not just for finite or discrete quantum systems [1] but also for continuum quantum field theories and holography [2,3]. In the continuum setting, however, entanglement entropies are notoriously hard to calculate and only few exact results exist. Lucky exceptions are provided by 2d conformal field theories [4] and the 2d free fermion [5,6]. The 2d free fermion system is a particularly fruitful playground since not only the entanglement entropy, but exact expressions for the modular Hamilotonian of multicomponent regions can be computed [6–8]. In particular, the recent papers [7,8] considered entanglement entropy and modular Hamiltonians for the 2d free fermion in the presence of a boundary or defect. In [8], the intervals that were considered were symmetric about a semitransparent defect that separates 2d flat space into two half-spaces.

We extend this repertoire of results by computing the entanglement entropy of 2d free fermions in the presence of a semitransparent defect, but allowing for the region under consideration to be asymmetric about the defect. Let $(x^0, x^1)$ be flat coordinates on 2d Minkowski space, and let the defect be located

at $x^1 = 0$. The $t$ be the transmission coefficient through the defect, and let $r = 1 - t$ be the reflection coefficient. Let us focus on a single interval $[-L_-, L_+]$ that straddles the defect (and so $L_-$ and $L_+$ are both positive real numbers). We are not able to compute the entanglement entropy of this interval in general, but we obtain results in three limits:

1. When the defect is almost completely transmitting, so that $r$ is small

2. When the defect is almost completely reflecting, so that $t$ is small

3. When the interval is almost symmetric about the defect, so that $\gamma = \frac{L_- - L_+}{L_- + L_+}$ is small.

These results are presented in section 2.4, see equations (2.41), (2.42) and (2.43). To the best of our knowledge, these formulas were not known before. In order to obtain these results, we have used the method of [5] and used the results in [9–11] for free fermion determinants in the presence of a boundary with non-translation invariant boundary conditions. This method can be used to obtain the entropies of any region, and most of our formal setup carries over to the more general case, but we focus on the particular application at hand to obtain explicit results.

Intuitively, the entropy should increase monotonically with the transmission coefficient $t$ since the defect acts as a coupling between left and right, and hence the increasing coupling increases the entanglement entropy. Our results (2.41), (2.42) and (2.43) are consistent with this intuitive behavior.

Earlier results on von Neumann entropy in the presence of interfaces include [12–14]. See section 3.2.1 of the review [4] for pointers to a few more results. See, for example, [15] for a study of the boundary entropy in holography.

## 1.2 Entanglement islands and graybody factors of black holes

Recent progress in the black hole information paradox has involved the semiclassical computation of the von Neumann entropy of Hawking radiation, reproducing the Page curve [16, 17]. The main player is the existence of a nontrivial Quantum Extremal Surface (QES) [18] at times larger than the Page time, whose generalized entropy tracks the shrinking area of the black hole horizon. The entanglement wedge of the Hawking radiation after the Page time contains a disconnected region deep in the gravitating spacetime, dubbed the entanglement island in [19].

Reference [20] exhibited the presence of entanglement islands in much simpler contexts that also play a role in avoiding an entropy paradox that exists in the eternal Schwarzschild geometry, due to Mathur [21]. By now, the presence of entanglement islands is well established in a wide variety of settings [22–40]. It has also been established that the nontrivial QES arises because spacetime wormholes dominate the computations of Rényi entropies [41, 42]. See the recent reviews [43, 44] for more on the black hole information problem and an overview of recent progress.

It is well-known that there are graybody factors in Hawking radiation [45]. It is hard to include graybody factors in the computations of the von Neumann entropy of bulk matter fields, and thus in

the setups of [17, 20], which couple a flat space bath to AdS space, the interface is taken to be fully transparent. This is rightly so, since graybody factors are not expected to affect the qualitative features of the Page curve. The paper [16] contained a qualitative discussion of graybody factors. The effect of graybody factors is also implicitly included in the higer-dimensional doubly-holographic setup of [22]. However, one would like to be more computationally explicit about graybody factors.

We take up this challenge in this paper in what is perhaps the simplest example of an entanglement island: the static zero temperature setup in section 2 of [20]. It was shown that when a zero-temperature AdS$_2$ black hole is coupled to flat, non-gravitating half-space via a fully transparent boundary, the QES of the boundary of AdS$_2$ does not lie at the Poincaré horizon, but lies at a finite value of the Poincaré radial coordinate. In other words, the entanglement wedge of the flat space region contains an entanglement island: the region between the Poincaré horizon and the QES.

In this work, instead of taking the AdS$_2$-flat space interface to be fully transparent, we take it to have a transmission coefficient $t$ and a reflection coefficient $r = 1 - t$. This is a toy model for graybody effects in the atmosphere of a black hole. In the fully transparent case, reference [20] found a nontrivial QES and an entanglement wedge. In the fully reflecting case, the QES is at the Poincaré horizon: this is easy to see, since the AdS$_2$ and flat space regions are not coupled at all in this case, the entanglement wedge of the boundary of AdS$_2$ better be the entire Poincaré patch of AdS$_2$.

We take the bulk matter to be 2d free fermions, and use our results (2.41), (2.42) and (2.43) for the entanglement entropy of free fermions in the presence of a defect to see how the QES moves as we vary the reflection coefficient. What we find is that if we perturb away from fully transmitting case, the QES moves towards the Poincaré horizon from its location in [20]. If we perturb away from the fully reflecting case, the QES moves from the Poincaré horizon towards the boundary of AdS$_2$.

In brief, our results support the hypothesis that the location of QES behaves monotonically and smoothly interpolates between its locations at $t = 1$ (the fully transmitting case) and $t = 0$ (the fully reflecting case).

As future work, it would be interesting to extend our results to the islands in the eternal Schwarzschild geometry at finite temperature [20] and also the evaporating case [16, 17]. We suspect that the location of the QES behaves monotonically with the strength of graybody effects in all these cases. Independently of the motivation from black hole physics, it would also be valuable to go beyond the limits we have considered obtain exact results for the 2d free fermion von Neumann entropy and the modular Hamiltonian for a general set of intervals in the presence of a defect.

In section 2, we outline the computation of the entanglement entropy of free fermions in the presence of a semitransparent interface and present the results. The details are contained in appendix A and B. In section 3, we recap the zero temperature entanglement island of [20] and show that the entanglement island behaves monotonically with the strength of graybody effects.

## 2 Entanglement entropy of free fermions with a semitransparent interface

The 2d massless Dirac fermion is a simple theory where one can explicitly compute not just the entanglement entropies of a region consisting of multiple intervals, but also the associated modular hamiltonians [5–8].

To study the entanglement problem in the presence of a defect, we take the fermions to live on the line, with the defect placed at $x^1 = 0$. We denote the region $x^1 > 0$ by $\Omega^+$ and the region $x^1 < 0$ by $\Omega^-$. Fields living in the respective half-planes will carry a $+$ or $-$ superscript. The defect is only partially transparent, with a transmission coefficient $t$. We will be more precise in specifying the boundary conditions below.

We take the $\gamma$-matrices to be in the Weyl basis,

$$\gamma^0_{\text{Lor}} = \begin{pmatrix} 0 & -i \\ -i & 0 \end{pmatrix}, \quad \gamma^1 = \begin{pmatrix} 0 & i \\ -i & 0 \end{pmatrix}, \quad \gamma_* = \gamma^0_{\text{Lor}} \gamma^1 = \begin{pmatrix} -1 & 0 \\ 0 & 1 \end{pmatrix}. \tag{2.1}$$

Here $\gamma_*$ denotes the chirality matrix and it is diagonal in the Weyl basis. The Dirac operator is

$$i\gamma^\mu_{\text{Lor}} \partial_\mu = \begin{pmatrix} 0 & \partial_0 - \partial_1 \\ \partial_0 + \partial_1 & 0 \end{pmatrix}. \tag{2.2}$$

The two-component fermions are taken to have components

$$\psi = \begin{pmatrix} \psi_R \\ \psi_L \end{pmatrix}. \tag{2.3}$$

The equation of motion says that $\psi_L$ is only a function of $x^0 + x^1$, which represents a left-moving wave, hence the subscript $L$ for $\psi_L$; a similar logic applies for $\psi_R$.

From now on, we work in Euclidean signature using the convention $x_2 = ix^0$. The ordering of the Euclidean coordinates will be $(x_1, x_2)$. For future use, we also note that

$$\gamma^2 = i\gamma^0_{\text{Lor}} = \begin{pmatrix} 0 & 1 \\ 1 & 0 \end{pmatrix}. \tag{2.4}$$

The boundary condition imposed at the defect is

$$\begin{pmatrix} \psi_R^+(0, x_2) \\ \psi_L^-(0, x_2) \end{pmatrix} = \mathbb{S} \begin{pmatrix} \psi_L^+(0, x_2) \\ \psi_R^-(0, x_2) \end{pmatrix}, \tag{2.5}$$

where $\mathbb{S}$ is the unitary scattering matrix

$$\mathbb{S} = \begin{pmatrix} c_1 & c_2 \\ c_3 & c_4 \end{pmatrix} = \begin{pmatrix} c_1 & -e^{i\phi} c_3^* \\ c_3 & e^{i\phi} c_1^* \end{pmatrix}. \tag{2.6}$$

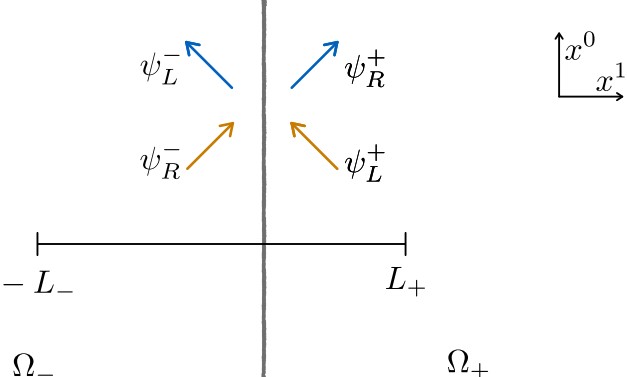

Figure 1: Setup for the fermion problem with a semitransparent interface. The interface or defect is located at $x_1 = 0$, and the regions $x_1 < 0$ and $x_1 > 0$ are denoted by $\Omega_-$ and $\Omega_+$, respectively. At $x_1 = 0$ the fermion fields are related by the $\mathcal{S}$-matrix in (2.5). For the entanglement computation we consider an interval $[-L_-, L_+]$ that straddles the defect at $x_1 = 0$.

These boundary conditions are energy conserving, and preserve one copy of Virasoro algebra after folding the plane along the defect. As can be seen in figure 1, the modes $\psi_R^+$ and $\psi_L^-$ are outgoing from the defect, whereas $\psi_L^+$ and $\psi_R^-$ are incoming. The quantity $t = |c_2|^2 = |c_3|^2$ is interpreted as a transmission coefficient, while $r = 1 - t = |c_1|^2 = |c_4|^2$ is interpreted as a reflection coefficient. The purely transmitting and the purely reflecting $\mathcal{S}$ matrices are

$$\mathcal{S}^{\mathrm{t}} = \begin{pmatrix} 0 & 1 \\ 1 & 0 \end{pmatrix}, \quad \mathcal{S}^{\mathrm{r}} = \begin{pmatrix} 1 & 0 \\ 0 & 1 \end{pmatrix}. \tag{2.7}$$

It will be helpful to rewrite the boundary condition (2.5) as

$$\mathcal{B}\psi(0, x_2) = 0, \tag{2.8}$$

where

$$\mathcal{B} = \begin{pmatrix} 1 & -c_1 & -c_2 & 0 \\ -c_1^* & 1 & 0 & -c_3^* \\ -c_2^* & 0 & 1 & -c_4^* \\ 0 & -c_3 & -c_4 & 1 \end{pmatrix}, \quad \psi(0, x_2) = \begin{pmatrix} \psi_R^+(0, x_2) \\ \psi_L^+(0, x_2) \\ \psi_R^-(0, x_2) \\ \psi_L^-(0, x_2) \end{pmatrix}. \tag{2.9}$$

Note that $\mathcal{B}$ is a Hermitian matrix with rank two.

We consider the fermions to be in their ground state and focus on the reduced density matrix $\rho_A$ of a given subset $A \subset \mathbb{R}$. We specify $A$ as a collection of disjoint intervals $[u_i, v_i]$ with $i \in \{1, \ldots, p\}$. We will mostly be interested in the case of a single interval $[-L_-, L_+]$ that straddles the defect (both $L_-$ and $L_+$ are positive real numbers). See figure 1 for the setup. We intend to compute the entanglement entropy $S(A)$ given by

$$S(A) = -\operatorname{Tr}(\rho_A \log \rho_A) = \lim_{n \to 1} S_n(A), \tag{2.10}$$

where $S_n(\mathcal{A})$ is the Rényi entropy,

$$S_n(\mathcal{A}) = \frac{1}{1-n} \log \mathrm{Tr}\left(\rho_{\mathcal{A}}^n\right). \tag{2.11}$$

## 2.1 Decoupling the replicas with gauge fields

We use the replica trick to compute $\mathrm{Tr}\left(\rho_{\mathcal{A}}^n\right)$. Our treatment closely follows [5]. The trace of $\rho_{\mathcal{A}}^n$ is given by the functional integral $Z[n]$ on the replica manifold,

$$\mathrm{Tr}\,\rho_{\mathcal{A}}^n = \frac{Z[n]}{Z[1]^n}, \tag{2.12}$$

where $Z[1]$ serves as a normalization factor that sets $\mathrm{Tr}\,\rho_{\mathcal{A}} = 1$.

When evaluating this trace for a fermionic field, we need to introduce a minus sign in the path integral boundary condition connecting the fields between the first and last replica copies [46]. Moreover, there is an additional factor of $-1$ for every copy because of non-trivial Lorentz rotation around the points $u_i$ and $v_i$ which is present in the Euclidean Hamiltonian when expressing $\mathrm{Tr}\,\rho_{\mathcal{A}}^n$ as a path integral [47]. Thus, we get an overall factor of $(-1)^{n+1}$ when connecting the fields along the first and last cuts.

Instead of dealing with the fields on the non-trivial replica manifold, we can work on a single plane by using the $n$-component field

$$\vec{\psi} = \begin{pmatrix} \psi_1(\mathbf{x}) \\ \vdots \\ \psi_n(\mathbf{x}) \end{pmatrix}, \tag{2.13}$$

where $\psi_j$ is the fermion field on the $j^{\mathrm{th}}$ sheet of the replica manifold. The vector $\vec{\psi}$ is not single valued: If we go in a circuit around $u_i$, the vector $\vec{\psi}$ gets multiplied by the matrix

$$T = \begin{pmatrix} 0 & 1 & 0 & \cdots & 0 & 0 \\ 0 & 0 & 1 & \cdots & 0 & 0 \\ \vdots & \vdots & \vdots & \ddots & \vdots & \vdots \\ 0 & 0 & 0 & \cdots & 0 & 1 \\ (-1)^{n+1} & 0 & 0 & \cdots & 0 & 0 \end{pmatrix}, \tag{2.14}$$

whereas if we go in a circuit around $v_i$, it gets multiplied by $T^{-1}$.

We can diagonalize $T$ by performing a unitary transformation. Note that the eigenvalues of $T$ are $e^{2\pi i k/n}$ with $k \in \{-\frac{n-1}{2}, -\frac{n-1}{2}+1, \ldots, \frac{n-1}{2}\}$. After this unitary transformation we end up with the decoupled fields $\psi_k$ living on a single plane. The fields $\psi_k$ are multivalued and get multiplied by $e^{2\pi i k/n}$ when encircling $u_i$, and by $e^{-2\pi i k/n}$ when encircling $v_i$. Note that this unitary transformation acts identically on the components of $\vec{\psi}$, so the boundary condition for $\psi_k$ is still

$$\mathcal{B}\psi_k(0, x_2) = 0. \tag{2.15}$$

We can get rid of the multivaluedness of $\psi_k$ by introducing an external gauge field $A_{k,\mu}(\mathbf{x})$. The gauge field is vortex-like near the end-points of the intervals:

$$\oint_{u_i} A_k = -\frac{2\pi k}{n} \,, \quad \oint_{v_i} A_k = \frac{2\pi k}{n} \,, \tag{2.16}$$

where the integrals are over closed contours encircling $u_i$ or $v_i$. These holonomies are captured by the field strength

$$F_{k,12}(\mathbf{x}) = -\frac{2\pi k}{n} \sum_{i=1}^{p} \left( \delta^{(2)}(\mathbf{x} - \mathbf{u}_i) - \delta^{(2)}(\mathbf{x} - \mathbf{v}_i) \right) \,. \tag{2.17}$$

where $\mathbf{u}_i = (u_i, 0)$ and $\mathbf{v}_i = (v_i, 0)$. Requiring that $A_k^\mu$ vanishes at infinity, we get the explicit formula

$$A_{k,\mu}(\mathbf{x}) = \frac{k}{n} \epsilon_{\mu\nu} \sum_{i=1}^{p} \left( \frac{(\mathbf{x} - \mathbf{u}_i)^\nu}{|\mathbf{x} - \mathbf{u}_i|^2} - \frac{(\mathbf{x} - \mathbf{v}_i)^\nu}{|\mathbf{x} - \mathbf{v}_i|^2} \right) \,, \tag{2.18}$$

where we use the standard $\epsilon_{12} = -\epsilon_{21} = 1$.

In presence of the gauge field, the fermion action is

$$I_k[\psi_k, \overline{\psi}_k; A_k] = \int_{\Omega^+} \mathrm{d}^2 x \, \overline{\psi}_k^+ \gamma^\mu \left( \partial_\mu + \mathrm{i} A_{k,\mu} \right) \psi_k^+ + \int_{\Omega^-} \mathrm{d}^2 x \, \overline{\psi}_k^- \gamma^\mu \left( \partial_\mu + \mathrm{i} A_{k,\mu} \right) \psi_k^- \,. \tag{2.19}$$

The functional integral factorizes into

$$Z[n] = \prod_{k=-(n-1)/2}^{(n-1)/2} Z_k \,, \tag{2.20}$$

where $Z_k$ is the functional integral over $\{\psi_k, \overline{\psi}_k\}$ with the background gauge field $A_k$ given by (2.18) and with boundary conditions (2.15); the matrix $\mathcal{B}$ is given in (2.9).

## 2.2  Computing the functional integral $Z_k$

In this section, we outline the calculation of the functional integral (the details are in the appendices)

$$Z_k = \int D\overline{\psi}_k^+ D\psi_k^+ D\overline{\psi}_k^- D\psi_k^- \exp\left( -\int_{\Omega^+} \mathrm{d}^2 x \, \overline{\psi}_k^+ \left( \slashed{\partial} + \mathrm{i}\slashed{A}_k \right) \psi_k^+ - \int_{\Omega^-} \mathrm{d}^2 x \, \overline{\psi}_k^- \left( \slashed{\partial} + \mathrm{i}\slashed{A}_k \right) \psi_k^- \right), \tag{2.21}$$

with the background gauge field $A_k$ given by (2.18) and with boundary conditions (2.15); the matrix $\mathcal{B}$ is given in (2.9). An essential fact is that the chiral anomaly in two dimensions completely determines the dependence of the functional integral on the gauge field [48].[1]

A general gauge field in two dimensions can be expressed as a sum of a gradient and a curl

$$A_{k,\mu} = \partial_\mu \eta_k - \epsilon_{\mu\nu} \, \partial_\nu \Phi_k \,. \tag{2.22}$$

For the gauge field profile in (2.18), we can choose

$$\eta_k(\mathbf{x}) = 0, \quad \Phi_k(\mathbf{x}) = -\frac{k}{n} \sum_{i=1}^{p} \log \frac{|\mathbf{x} - \mathbf{u}_i|}{|\mathbf{x} - \mathbf{v}_i|} \,. \tag{2.23}$$

---

[1] An equivalent method is to bosonize the fermions.

Note that $\Phi_k \sim \frac{1}{|\mathbf{x}|}$ as $|\mathbf{x}| \to \infty$.[2] Given a background gauge field of the form (2.22), we can decouple the fermions $\psi_k^+$ from the background gauge field $A_k$ by a change of variables which is a combination of gauge and chiral transformations

$$\psi_k^+ = \exp\left(i\eta_k + \gamma_* \Phi_k\right) \chi_k^+ \,. \tag{2.24}$$

Essentially, we are changing variables from $\psi_k^+$ to $\chi_k^+$ in the functional integral. The change in the fermionic measure under this transformation is nontrivial

$$D\overline{\psi}_k^+ D\psi_k^+ = J_k^+ \, D\overline{\chi}_k^+ D\chi_k^+ \,, \tag{2.25}$$

where $J_k^+$ is the Jacobian of the transformation. For fermions living on the full line, the result for this Jacobian is well-known [48]. In case of manifolds with boundaries, we need to be careful about possible boundary contributions.

To separate the bulk and boundary contributions, we divide the positive real line into two intervals $(0, \epsilon^+)$ and $(\epsilon^+, \infty)$. For the interval $(\epsilon^+, \infty)$, we can obtain the bulk contribution using the result for closed manifolds [48],

$$\left(J_k^+\right)_{\text{bulk}} = \exp\left(-\frac{1}{2\pi} \int_{\Omega^+(\epsilon^+)} \mathrm{d}^2 x \, \partial_\mu \Phi_k(\mathbf{x}) \partial^\mu \Phi_k(\mathbf{x})\right) \,, \tag{2.26}$$

where $\Omega^+(\epsilon^+) = \{(x_1, x_2) : x_1 > \epsilon^+\}$. The boundary contribution is [11]

$$\left(J_k^+\right)_{\text{bdry}} = \exp\left(-\frac{1}{4\epsilon^+} \int_{\mathbb{R}} \mathrm{d}x_2 \, \Phi_k(0, x_2)\right) \,. \tag{2.27}$$

A similar treatment can be done for the fermions on $\Omega_-$ to get the full Jacobian

$$J_k = \exp\left(-\frac{1}{2\pi} \int_{\Omega^+(\epsilon^+) \cup \Omega^-(\epsilon^-)} \mathrm{d}^2 x \, \partial_\mu \Phi_k(\mathbf{x}) \partial^\mu \Phi_k(\mathbf{x}) - \left(\frac{1}{4\epsilon^+} + \frac{1}{4\epsilon^-}\right) \int_{\mathbb{R}} \mathrm{d}x_2 \, \Phi_k(0, x_2)\right) \,, \tag{2.28}$$

with $\epsilon^- < 0$. Once we do the sum over $k$, the boundary terms will vanish (since $\Phi_k$ is linear in $k$) and henceforth they will be omitted.[3] Taking the limit $\epsilon^+, \epsilon^- \to 0$, we get

$$J_k = \exp\left(-\frac{1}{2\pi} \int_\Omega \mathrm{d}^2 x \, \partial_\mu \Phi_k \, \partial^\mu \Phi_k\right) \,. \tag{2.29}$$

This is the result for the Jacobian on the entire plane in [48].

The integral in (2.29) is easily evaluated by substituting $\Phi_k$ from (2.23), integrating by parts once and using the fact that $\nabla^2 \Phi$ is a sum of delta functions. The result is

$$J_k = \exp\left(-\frac{2k^2}{n^2} \Xi(\{u_i\}, \{v_j\})\right) \,, \tag{2.30}$$

---

[2] This decomposition of the gauge field as the sum of a gradient and a curl is not unique. In Appendix B, we use a different decomposition which allows us to easily calculate the results in the purely reflecting case.

[3] We can also choose $\epsilon^+ = -\epsilon^-$, and then this term automatically vanishes.

where [5]

$$\Xi(\{u_i\}, \{v_j\}) := \sum_{i,j} \log|u_i - v_j| - \sum_{i<j} \log|u_i - u_j| - \sum_{i<j} \log|v_i - v_j| - p\log\varepsilon \,. \tag{2.31}$$

Here, we have introduced the short-distance cutoff $\varepsilon$ to split the coincidence points $|u_i - u_i|, |v_i - v_i| \to \varepsilon$, and the sum over $i, j$ is over all the intervals comprising the region $\mathcal{A}$.

So far, we changed variables from $\psi_k$ to $\chi_k$ (2.24) in the functional integral and obtained the Jacobian for this transformation. Since the Jacobian (2.30) is independent of $\chi_k$, the functional integral can be expressed as

$$Z_k = J_k \widetilde{Z}_k \,, \tag{2.32}$$

where $\widetilde{Z}_k$ is the path integral over $\chi_k$, now without any gauge fields

$$\widetilde{Z}_k = \int D\overline{\chi}_k^+ D\chi_k^+ D\overline{\chi}_k^- D\chi_k^- \exp\left(-\int_{\Omega^+} \mathrm{d}^2x \, \overline{\chi}_k^+ \slashed{\partial} \chi_k^+ - \int_{\Omega^-} \mathrm{d}^2x \, \overline{\chi}_k^- \slashed{\partial} \chi_k^-\right). \tag{2.33}$$

This integral gives a nontrivial contribution to the partition function (that depends on the interval endpoints $u_i$ and $v_i$) because of the boundary conditions obeyed by $\chi_k$. Using the definition of $\chi_k$ in (2.24) and the boundary condition (2.15) for $\psi_k$, we see that the boundary condition for $\chi_k$ is

$$\mathcal{B}\begin{pmatrix} e^{\gamma_* \Phi_k(0,x_2)} & 0 \\ 0 & e^{\gamma_* \Phi_k(0,x_2)} \end{pmatrix} \chi_k(0, x_2) = 0 \,, \tag{2.34}$$

where we have combined the two-component objects $\chi_k^+$ and $\chi_k^-$ into a four component object $\chi_k$ along the lines of the second relation in (2.9). We can rewrite this boundary condition as

$$\mathcal{B}\begin{pmatrix} e^{H_k(x_2)} & 0 & 0 & 0 \\ 0 & 1 & 0 & 0 \\ 0 & 0 & e^{H_k(x_2)} & 0 \\ 0 & 0 & 0 & 1 \end{pmatrix} \chi_k(0, x_2) = 0, \quad \text{with } H_k(x_2) := -2\Phi_k(0, x_2) \,. \tag{2.35}$$

The computation of the functional integral (2.33) subject to the boundary condition (2.35) is given in Appendix A. The main idea is to use a theorem due to Forman [9], which was also used in [10, 11], in order to relate the fermion determinant with the position-dependent boundary condition (2.35) to the fermion determinant with the much simpler boundary condition $\mathcal{B}\chi_k(0, x_2) = 0$. In the process, one still needs to compute the trace of an infinite matrix, and we have been unable to solve this problem in general. We have however been able to obtain results in the three limits described in the introduction. We will present our new results in section 2.4, but let us first review the known results about the fully transmitting and the fully reflecting cases.

## 2.3  Review: The purely transmitting and the purely reflecting cases

**The purely transmitting case.** In the purely transmitting case corresponding to $\mathcal{S}^{\mathrm{t}}$ in (2.7), the boundary condition in (2.35) is equivalent to $\mathcal{B}\chi_k(0, x_2) = 0$, so $\widetilde{Z}_k = Z[1]$. Thus, the entropies come

purely from the Jacobians $J_k$ given in (2.30) [5]. The result for the von Neumann entropy is

$$S(\mathcal{A}) = \frac{1}{3} \, \Xi(\{u_i\}, \{v_j\}) \qquad \text{(fully transmitting)}, \tag{2.36}$$

where $\Xi$ was defined in (2.31). This is just the result of [5]. For a single interval $[-L_-, L_+]$, the answer has the familiar logarithmic expression dependence,

$$S([-L_-, L_+]) = \frac{1}{3} \log \frac{L_+ + L_-}{\varepsilon} \qquad \text{(fully transmitting)}. \tag{2.37}$$

**The purely reflecting case.** In the purely reflecting case corresponding to $\mathcal{S}^{\mathrm{r}}$ in (2.7), we can compute the functional integral by using a different decomposition for the gauge field in terms of $\eta_k$ and $\Phi_k$. This calculation is done in Appendix B. If none of the intervals intersects the defect, we get

$$S(\mathcal{A}) = \frac{1}{3} \xi \left( \{u_i^+\}, \{v_j^+\} \right) + \frac{1}{3} \xi \left( \{u_i^-\}, \{v_j^-\} \right) \qquad \text{(fully reflecting)}, \tag{2.38}$$

with $\xi$ defined as

$$\xi \left( \{u_i\}, \{v_j\} \right) := \Xi \left( \{u_i\}, \{v_j\} \right) - \frac{1}{2} \sum_{i,j} \log \left( \frac{|u_i + v_j||v_i + u_j|}{|u_i + u_j||v_i + v_j|} \right). \tag{2.39}$$

Here, $u_i^+$, $v_i^+$ refer to the endpoints in the right half-plane $\Omega^+$, and $u_i^-$, $v_i^-$ refer to the endpoints in the left half-plane $\Omega^-$. For a single interval on one side of the boundary, one can check that the the reflecting answer reduces to the transmitting answer if the interval is far from the boundary.

For the interval $[-L_-, L_+]$, which is our case of interest, the entropy is

$$S([-L_-, L_+]) = \frac{1}{6} \log \frac{2L_-}{\varepsilon} + \frac{1}{6} \log \frac{2L_+}{\varepsilon} \qquad \text{(fully reflecting)}. \tag{2.40}$$

Notice that the entropy in the fully reflecting case is a sum of entropies on the left and right side of the defect, since the two half-planes are completely decoupled.

We now turn to our results for the entropy with nonzero reflection and transmission coefficients.

## 2.4 Results for von Neumann entropy

We will consider entanglement entropies for the case of a single interval $[-L_-, L_+]$ that straddles the defect[4] (so we have $L_- > 0$ and $L_+ > 0$), since this is what will be important for the application in section 3. The details can be found in the appendices.

1. When the defect is almost fully transmitting, so that $r$ is small, we get

$$S([-L_-, L_+]) = \frac{1}{3} \log \frac{L_+ + L_-}{\varepsilon} + \frac{r}{8} \left( 1 + \frac{L_-^2 + L_+^2}{L_-^2 - L_+^2} \log \frac{L_+}{L_-} \right) + O(r^2). \tag{2.41}$$

---

[4] The entanglement entropies for the interval $[L_-, L_+]$ (with $L_- > 0$) can be obtained from the following results using $S([L_-, L_+]) = S([-L_-, L_+]) + \frac{1}{3} \log \frac{L_+ - L_-}{L_+ + L_-}$. This relationship follows because the contribution to the entropy coming from $\widetilde{Z}_k$ are equal in the two cases. That happens because $\Phi_k$ on the boundary is the same in both cases.

2. When the defect is almost fully reflecting, so that $t$ is small, we get

$$S([-L_-, L_+]) = \frac{1}{6} \log \frac{4L_+L_-}{\varepsilon^2} + \frac{t}{8} \left( 1 + \frac{L_-^2 - 6L_-L_+ + L_+^2}{2(L_- - L_+)\sqrt{L_-L_+}} \tan^{-1} \left( \frac{L_- - L_+}{2\sqrt{L_-L_+}} \right) \right) + O(t^2).$$

(2.42)

3. Keeping $r$ and $t$ general, but assuming that $\gamma = \frac{L_- - L_+}{L_- + L_+}$ is small, we get

$$S([-L_-, L_+]) = \frac{1}{3} \log \frac{L_+ + L_-}{\varepsilon} - \frac{r}{6} \left( \gamma^2 + \frac{\gamma^4}{2} + \frac{\gamma^6}{3} + \frac{\gamma^8}{4} \right) + \frac{rt}{30} \left( \frac{\gamma^4}{2} + \frac{\gamma^6}{2} + \frac{65\gamma^8}{144} \right)$$
$$+ \frac{rt}{42} \left[ (r - t) \left( \frac{\gamma^6}{6} + \frac{\gamma^8}{4} \right) - \frac{\gamma^8}{72} \right] + \frac{rt(r - 5t)(5r - t)}{30} \left( \frac{\gamma^8}{144} \right) + O\left(\gamma^{10}\right). \quad (2.43)$$

Note that the infinite series in $\gamma$ multiplying $r$ resums to $-\frac{1}{6} \log(1 - \gamma^2) = -\frac{1}{6} \log \frac{4L_-L_+}{(L_+ + L_-)^2}$, which, when $r = 1$, converts the purely transmitting result $\frac{1}{3} \log \frac{L_+ + L_-}{\varepsilon}$ to the fully reflecting result $\frac{1}{6} \log \frac{4L_+L_-}{\varepsilon^2}$. All other terms are proportional to $rt$ and thus vanish at the two extremes.

It is also straightforward to check that the $O(r)$ term in (2.41) and the $O(t)$ term in (2.42) vanish when $\gamma = 0$, reproducing the result in [8] that the entropy of a symmetrically located interval is not affected by the reflection and transmission coefficients.[5] Furthermore, the $O(r)$ term in (2.41) is always negative and the $O(t)$ term in (2.42) is always positive, supporting the intuitive results that the entanglement is the largest when the defect is fully transmitting.

As future work, it would be interesting to determine exact entanglement entropy (and, if possible, the modular Hamiltonian) outside the various limits that we have considered.

# 3   Entanglement islands with graybody factors

In this section, we will apply our results for the von Neumann entropy to the zero temperature entanglement island calculations of [20]. The new component is the introduction of a reflection/transmission coefficient at the interface between AdS$_2$ and flat space. We will first quickly review the setup (see figure 2) and then show that the entanglement island behaves in a monotonic fashion as a function of the reflection/transmission coefficient.

The gravitational theory is taken to be AdS-JT gravity coupled to matter, whose action is given by [49]

$$I[g, \phi, \psi] = \phi_0 + \frac{1}{4\pi} \int \mathrm{d}^2x \sqrt{g}\, \phi\, (R + 2) + I_{\mathrm{CFT}}[g, \psi], \qquad (3.1)$$

where we have put $4G_N = 1$, the constant $\phi_0$ gives the extremal entropy, and we have omitted the Gibbons-Hawking boundary term. The CFT matter action is given by $I_{\mathrm{CFT}}$ and we take the matter

---

[5] More generally, this happens since the $\Phi_k$ in (2.23) vanishes on the interface when the region $\mathcal{A}$ is symmetrically placed with respect to the interface.

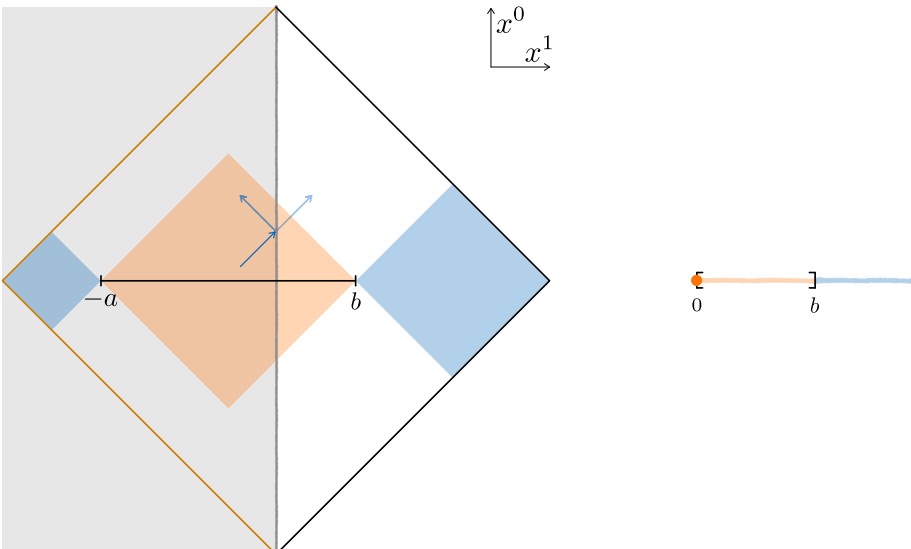

Figure 2: **Left**: Setup for the QES calculation for a partially reflecting AdS$_2$ boundary. In the gray region gravity is dynamical and we will consider the Poincaré patch with its horizon drawn in orange. This region is glued along a the AdS$_2$ boundary to a flat space region on the right. The free fermion matter lives in both the AdS$_2$ and flat space regions, but there is partial reflection at the interface, with transmission coefficient $t$ (and reflection coefficient $r = 1 - t$). This is a toy model to capture the graybody effects in the black hole atmosphere. A candidate entanglement wedge $[-a, b]$ is shaded orange and its complement in shaded blue. **Right**: The dual SYK + wire system. The fundamental computation we are doing is of the entropy and the entanglement wedge of the interval $[\mathbf{0}, \mathbf{b}]$ in this non-gravitational description.

fields $\psi$ to not couple to the dilaton. For the gravity variables, we put the usual boundary conditions $g_{uu}|_{\text{bdy}} = 1/\varepsilon^2$ and $\phi|_{\text{bdy}} = \phi_r/\varepsilon$.

Using coordinates $(x^0, x^1)$ with $x^1 < 0$, the zero temperature solution of JT gravity is given by

$$\mathrm{d}s^2 = \frac{-(\mathrm{d}x^0)^2 + (\mathrm{d}x^1)^2}{(x^1)^2}, \quad \phi = \frac{\phi_r}{-x^1} \, . \tag{3.2}$$

The JT boundary conditions imply that the AdS$_2$ boundary is at $x^1 = -\varepsilon$. We now couple the AdS$_2$ region to half of 2d flat-space, which is non-gravitating. The matter fields are taken to be free fermions and they propagate on both the AdS and the flat space regions, but we choose the boundary conditions for them so that they see a partially transmitting interface at $x_1 = -\varepsilon$, see equation (2.5). This is how our work differs from the previous papers [17, 20].

As emphasized in [20], one should think of the entropies as being fundamentally defined in a candidate dual description that does not involve gravity. This will look something like an SYK model coupled to a wire. The transmission coefficient encodes some property of the coupling between the SYK model and the wire, and the zero temperature equilibrium geometry corresponds to the ground state of the coupled Hamiltonian.

The goal is to compute the entanglement wedge of the region $[\mathbf{0}, \mathbf{b}]$ in the SYK+wire description. A

candidate for a QES is the point $(x^0, x^1) = (0, -a)$ in the AdS$_2$ region (with $a > 0$). The entanglement wedge of $[\mathbf{0}, \mathbf{b}]$ will be the region $(-a_*, b)$ where $a_*$ is the location of the QES. We need to compute the generalized entropy functional $S_{\text{gen}}$ of a region $[-a, b]$ that is partially in the bath region and partially in the gravity region, see figure 2. For this, we need the dilaton profile given in (3.2) and also an expression for the entropy of bulk matter fields in the semiclassical description, which were obtained in the previous section.

The generalized entropy of the interval $[-a, b]$ is given by

$$S_{\text{gen}}(a) = \phi_0 + \frac{\phi_r}{a} + c\, S_{\text{Dirac, flat}}([-a, b]) - \frac{c}{6} \log a \,, \tag{3.3}$$

where the second term corresponds to the entropies we computed in section 2 and the final term comes from the Weyl factor in the metric (3.2) at the left end-point of the interval $[-a, b]$.[6] We have also taken $c$ copies of the free Dirac fermion system, and as usual $c$ is taken to be large so that the entropy coming from fluctuations in the gravity sector can be ignored. Let us define a quantity $k$ with dimensions of length as

$$k := \frac{6\phi_r}{c} \,. \tag{3.4}$$

Extremizing $S_{\text{gen}}(a)$ with respect to $a$ gives us the location $a_*$ of the QES. As before, we specialize to three cases:

1. When the reflection coefficient $r$ is small, we use (2.41) for $S_{\text{Dirac, flat}}$ and we get

$$a_*(r) = a_0 + r \frac{3a_0^2 \left(a_0^4 - b^4 - 4a_0^2 b^2 \log(a_0/b)\right)}{4(a_0 - b)^2 \left((a_0 + b)^2(a_0 + 2k) - 2a_0^3\right)} + O(r^2) \,, \tag{3.5}$$

where $a_0$ is the location of the island in the purely transmitting case [20],

$$a_0 = a_*(r = 0) = \frac{1}{2} \left(b + k + \sqrt{b^2 + 6bk + k^2}\right) \,. \tag{3.6}$$

It is also instructive to look at the extreme limits for $b$ in this expression.

$$a_\star = \frac{6\phi_r}{c} + r \frac{9\phi_r}{2c} + O(b, r^2) \qquad \text{(small } b)\,, \tag{3.7}$$

$$a_\star = \left(b + \frac{12\phi_r}{c}\right) + r \frac{12\phi_r}{c} + O(b^{-1}, r^2) \quad \text{(large } b)\,. \tag{3.8}$$

We would like to argue that the second term in (3.5) in always positive, so that the QES moves from its location $a_0$ in the fully transmitting case towards the Poincaré horizon, where it lies in the fully reflecting case. It is easy to plot the second term of (3.5) as a function of $b$ and $k$ and check that it is positive, but we would like to give a more illuminating argument. First, note that $a_0 > b$, which as observed in [20] is a condition that is imposed by the Quantum Focusing

---

[6] As already noted, the boundary conditions on the matter fields preserve one copy of the Virasoro algebra.

Conjecture [50]. The function $S'_\text{gen}(a)$ at full transmission is positive to the left of the QES, and negative to the right of the QES. We noted in section 2.4 that the correction term in (2.41) is negative definite and has a maximum value of 0 when $L_- = L_+$, or $a = b$ in the notation of this section. These facts dictate that the QES can only move leftwards from $a_0$.

2. When the transmission coefficient $t$ is small, we use (2.42) for $S_{\text{Dirac, flat}}$ and we get

$$a_*(r) = \left(\frac{1024\, bk^2}{9\pi^2}\right)^{\frac{1}{3}} t^{-\frac{2}{3}} + O(t^0)\,. \tag{3.9}$$

The QES goes to infinity as $t \to 0$. In the fully reflecting case, the QES is at the Poincaré horizon: the AdS$_2$ and flat space regions are not coupled at all in this case, and so the entanglement wedge of the boundary of AdS$_2$ better be the entire Poincaré patch of AdS$_2$.

Note that since $a_*$ is getting large, we might worry that the coefficient of the $O(t)$ term in the bulk entropy formula (2.42) is getting large. However, the overall size of the $O(t)$ term at the extremization point is seen to be $t^{\frac{2}{3}}$, which is small. In general, entropies cannot grow faster than linearly in the interval size, and so, if the expansion in $t$ is well-behaved, we do not expect large powers of $L_-$ to show up at higher orders in (2.42).

3. Taking $b$ to be very large, we expect, as in [20], that the quantity $\frac{a_*-b}{a_*+b}$ is small. Keeping terms up to order $\gamma^2$ in (2.43), we find

$$a_*(r) = b\left(1 + \frac{1}{1-r}\frac{2k}{b} + O\left(\frac{k^2}{b^2}\right)\right)\,. \tag{3.10}$$

This matches with the small $r$ result (3.5) in the common domain of validity, but we should not trust the exact form of this answer in the $r \to 1$ limit, since $\gamma$ is getting large in that limit.

To summarize, within the particular setup and the various limits that we have considered, we have shown that the location of the QES (and thus the size of the entanglement island) is a monotonic function of the transmission strength of the interface between the gravitating region and faraway flat space region. It would be interesting to extend our results to various other setups in which islands are known to exist. Our prejudice is that such monotonicity should always hold. In the case of an evaporating black hole, one should see the QES move from inside the horizon towards to horizon as one turns on graybody effects.

### Acknowledgments

We would like to thank Juan Maldacena for suggesting this problem. JK is supported by the Simons Foundation. RM is supported in part by Simons Investigator Award #620869. CM is supported in part by the U.S. Department of Energy under award DE-SC0019380.

# A Dirac determinant on the half-plane with non-translation invariant boundary conditions

In this appendix, we will compute the functional integral

$$\widetilde{Z} = \int D\overline{\chi}^+ D\chi^+ D\overline{\chi}^- D\chi^- \, \exp\left(-\int_{\Omega^+} \mathrm{d}^2 x \, \overline{\chi}^+ \slashed{\partial} \chi^+ - \int_{\Omega^-} \mathrm{d}^2 x \, \overline{\chi}^- \slashed{\partial} \chi^-\right). \tag{A.1}$$

where the fermionic fields satisfy the boundary conditions

$$\mathcal{B} \, e^{\mathcal{H}(x_2)} \, \chi(0, x_2) = 0 \,, \tag{A.2}$$

with the matrices

$$\mathcal{B} = \begin{pmatrix} 1 & -c_1 & -c_2 & 0 \\ -c_1^* & 1 & 0 & -c_3^* \\ -c_2^* & 0 & 1 & -c_4^* \\ 0 & -c_3 & -c_4 & 1 \end{pmatrix}, \quad \mathcal{H}(x_2) = \begin{pmatrix} H(x_2) & 0 & 0 & 0 \\ 0 & 0 & 0 & 0 \\ 0 & 0 & H(x_2) & 0 \\ 0 & 0 & 0 & 0 \end{pmatrix}, \tag{A.3}$$

and the vector $\chi$ will be specified shortly. The matrix $\mathcal{B}$ has rank two. We fold the left half plane, $\Omega_-$, to the right half-plane, $\Omega_+$. This folding changes the coordinate $\mathbf{x} = (x_1, x_2) \in \Omega_-$ to $\overline{\mathbf{x}} := (-x_1, x_2) \in \Omega_+$, so we have the transformation for derivatives $(\partial_1, \partial_2) \to (\overline{\partial_1}, \overline{\partial_2}) = (-\partial_1, \partial_2)$ and for the fermionic fields $\chi_R^-(\mathbf{x}) \to \chi_L^-(\overline{\mathbf{x}})$, $\chi_L^-(\mathbf{x}) \to \chi_R^-(\overline{\mathbf{x}})$. In this folded geometry, the Dirac operator is $i\slashed{\partial} \oplus i\overline{\slashed{\partial}}$. In contrast to (2.9), after the folding, the fermion field has components in the following order:

$$\chi = \begin{pmatrix} \chi_R^+ \\ \chi_L^+ \\ \chi_L^- \\ \chi_R^- \end{pmatrix} \tag{A.4}$$

with all fields having a position argument belonging to the right half plane $\Omega_+$.

We also remind the reader that for the application needed in the main text, $H_k(x_2) = -2\Phi_k(0, x_2)$ and $\Phi_k(\mathbf{x})$ is given in (2.23). The main tools needed to compute the partition function are described in references [9–11].

The fermion path integral (A.1) can be written as the functional determinant $\det(i\slashed{\partial} \oplus i\overline{\slashed{\partial}})$, subject to the boundary condition (A.2). To compute this determinant we will employ a theorem by Forman [9]. The idea is to consider a one-parameter family of boundary conditions labelled by $\tau$ such that $\tau = 0$ corresponds to the boundary condition $\mathcal{B}\chi(0, x_2) = 0$, and $\tau = 1$ corresponds to the boundary condition (A.2) that we are interested in. This gives rise to a family of functional determinants

$$\widetilde{Z}(\tau) = \det(i\slashed{\partial} \oplus i\overline{\slashed{\partial}})_{\mathcal{B}\mathcal{U}(\tau)}, \quad \mathcal{U}(\tau) := e^{\tau \mathcal{H}(x_2)}. \tag{A.5}$$

where the subscript labels the modified boundary condition $\mathcal{B}\mathcal{U}(\tau)\chi = \mathcal{B}e^{\tau\mathcal{H}(x_2)}\chi = 0$. Forman's

theorem (theorem 2 of [9], see also [10]) states that[7]

$$\frac{\mathrm{d}}{\mathrm{d}\mu} \log \frac{\widetilde{Z}(\tau + \mu)}{\widetilde{Z}(\mu)} = \frac{\mathrm{d}}{\mathrm{d}\mu} \log \det \Phi_\mu(\tau) .$$ (A.6)

We will explain the definition of the operator $\Phi$ below.

The Dirac operator $i\slashed{\partial} \oplus i\overline{\slashed{\partial}}$ is given in the Weyl basis ((2.1) and (2.4)) by

$$i\slashed{\partial} \oplus i\overline{\slashed{\partial}} = \begin{pmatrix} 0 & -\partial_1 + i\partial_2 & 0 & 0 \\ \partial_1 + i\partial_2 & 0 & 0 & 0 \\ 0 & 0 & 0 & \partial_1 + i\partial_2 \\ 0 & 0 & -\partial_1 + i\partial_2 & 0 \end{pmatrix} .$$ (A.7)

We will denote the elements of the kernel of this operator by $\chi$ and we take them to satisfy a fake boundary condition on the far-right side of the half-plane

$$\mathcal{B}\chi(L, x_2) = 0,$$ (A.8)

with $L \gg 1$. This boundary condition is arbitrary and is chosen for convenience, following [10]. Notice that we maintain translation symmetry along the line $x_1 = L$ in this fake problem (in contrast to the actual boundary condition (A.2) where the function $H(x_2)$ explicitly depends on $x_2$). We further compactify $x_2$ on a large circle of length $T$, and impose antiperiodic boundary conditions $\chi(x_1, -T/2) = -\chi(x_1, T/2)$. The elements of the kernel of (A.7) satisfying the boundary condition (A.8) and having definite Matsubara frequencies $w_n = (2n + 1)\pi/T$ with $n \in \mathbb{Z}$ can be easily found

$$\chi_{An}(x_1, x_2) = \frac{e^{-iw_n x_2}}{\sqrt{2\cosh(2w_n L)}} \begin{pmatrix} e^{-w_n(x_1-L)} \\ c_1^* e^{w_n(x_1-L)} \\ c_2^* e^{w_n(x_1-L)} \\ 0 \end{pmatrix}, \quad \chi_{Bn}(x_1, x_2) = \frac{e^{-iw_n x_2}}{\sqrt{2\cosh(2w_n L)}} \begin{pmatrix} 0 \\ c_3^* e^{w_n(x_1-L)} \\ c_4^* e^{w_n(x_1-L)} \\ e^{-w_n(x_1-L)} \end{pmatrix}.$$ (A.9)

Note that $\{\chi_{An}, \chi_{Bn}\}$ satisfy the orthonormality condition $\frac{1}{T}\int_{-T/2}^{T/2} \mathrm{d}x_2\, \chi_I^\dagger(0, x_2)\chi_J(0, x_2) = \delta_{IJ}$, with $I \in \{A, B\} \times \mathbb{Z}$. The vectors in (A.9) when evaluated on the true boundary $x_1 = 0$ will not satisfy the boundary condition (A.2). Now we ask the question: By how much do $\chi_I(0, x_2)$ fail to satisfy the true boundary condition (A.2)? This will be proportional to $\mathcal{B}\mathcal{U}(\tau)\chi_I(0, x_2)$. We now pick a fixed basis of functions on the line $x_1 = 0$ that also satisfy $\mathcal{B}\widetilde{\chi} = 2\widetilde{\chi}$.[8] Explicitly, these are

$$\widetilde{\chi}_{An}(x_2) = \frac{e^{-iw_n x_2}}{\sqrt{2}} \begin{pmatrix} 1 \\ -c_1^* \\ -c_2^* \\ 0 \end{pmatrix}, \qquad \widetilde{\chi}_{Bn}(x_2) = \frac{e^{-iw_n x_2}}{\sqrt{2}} \begin{pmatrix} 0 \\ -c_3^* \\ -c_4^* \\ 1 \end{pmatrix}.$$ (A.10)

---

[7] Please note that the operator $\Phi_\mu(\tau)$ has nothing to do with the function $\Phi_k(\mathbf{x})$.

[8] Since the vector $\mathcal{B}\mathcal{U}(\tau)\chi_I(0, x_2)$ is of the form $\mathcal{B}(\cdot)$, and $\mathcal{B}$ has eigenvalues 0 and 2, it lies in the subspace that has eigenvalue 2 under $\mathcal{B}$. Thus the inner product of $\mathcal{B}\mathcal{U}(\tau)\chi_I(0, x_2)$ and $\widetilde{\chi}_J$ measures the amount by which the boundary condition is violated.

With the two sets of vectors defined in (A.9) and (A.10), the matrix $h(\tau)$ is defined to have matrix elements

$$h_{IJ}(\tau) := \frac{1}{T} \int_{-T/2}^{T/2} \mathrm{d}x_2 \, \widetilde{\chi}_I^\dagger(x_2) \, \mathcal{B} \, e^{\tau \mathcal{H}(x_2)} \chi_J(0, x_2) \,, \tag{A.11}$$

with $I, J \in \{A, B\} \times \mathbb{Z}$. The intuition is that the matrix $h$ measures the failure of $\chi_J(x_1, x_2)$ to satisfy the correct boundary condition (A.2) for our problem. The quantity $\Phi_\mu(\tau)$ in (A.6) is defined via [9–11]

$$h(\mu + \tau) = \Phi_\mu(\tau) \, h(\mu) \,. \tag{A.12}$$

Our goal is to calculate $\widetilde{Z}$ at $\tau = 1$. Let us take a $\tau$ derivative of Forman's result, (A.6),

$$\frac{\mathrm{d}}{\mathrm{d}\mu} \frac{\mathrm{d}}{\mathrm{d}\tau} \log \frac{\widetilde{Z}(\mu + \tau)}{\widetilde{Z}(\mu)} = \frac{\mathrm{d}}{\mathrm{d}\mu} \frac{\mathrm{d}}{\mathrm{d}\tau} \mathrm{Tr} \log \left( h(\mu + \tau) h^{-1}(\mu) \right) \,. \tag{A.13}$$

On the LHS the $\tau$ derivative kills $\log \widetilde{Z}(\mu)$, whereas the RHS can be simplified by using $\frac{\mathrm{d}}{\mathrm{d}\tau} \mathrm{Tr} \log A(\tau) = \mathrm{Tr}(\dot{A}(\tau) \, A^{-1}(\tau))$ and cyclicity of the trace. So (A.13) simplifies to

$$\frac{\mathrm{d}^2}{\mathrm{d}\tau \mathrm{d}\mu} \widetilde{Z}(\mu + \tau) = \frac{\mathrm{d}}{\mathrm{d}\mu} \mathrm{Tr} \left( \frac{\mathrm{d}h(\mu + \tau)}{\mathrm{d}\tau} h^{-1}(\mu + \tau) \right) \,. \tag{A.14}$$

Notice that the $\mu$ derivatives can now be traded for $\tau$ derivatives and, after setting $\mu = 0$, we end up with the differential equation

$$\frac{\mathrm{d}^2}{\mathrm{d}\tau^2} \log \widetilde{Z}(\tau) = \frac{\mathrm{d}}{\mathrm{d}\tau} \mathrm{Tr} \left( \frac{\mathrm{d}h(\tau)}{\mathrm{d}\tau} h(\tau)^{-1} \right) \,. \tag{A.15}$$

To solve for $\widetilde{Z}(\tau)$, we integrate this twice and hence we need the value of $\widetilde{Z}'(\tau = 0)$. We claim that $\widetilde{Z}(\tau)$ is an even function of $\tau$ so $\widetilde{Z}'(\tau = 0) = 0$. This is because the boundary condition for $\widetilde{Z}(-\tau)$ is equivalent to

$$\mathcal{B} \begin{pmatrix} 1 & 0 & 0 & 0 \\ 0 & e^{\tau H(x_2)} & 0 & 0 \\ 0 & 0 & 1 & 0 \\ 0 & 0 & 0 & e^{\tau H(x_2)} \end{pmatrix} \chi(0, x_2) = 0 \,, \tag{A.16}$$

upto an overall phase factor. This can be converted to the boundary condition for $\widetilde{Z}(\tau)$ by interchanging the fermionic fields $\chi^+ \leftrightarrow \chi^-$. The action remains unchanged under this interchange, so $\widetilde{Z}(-\tau) = \widetilde{Z}(\tau)$. Hence we find,

$$\log \left( \frac{\widetilde{Z}(1)}{\widetilde{Z}(0)} \right) = \int_0^1 \mathrm{d}\tau \, \mathrm{Tr} \left( \frac{\mathrm{d}h(\tau)}{\mathrm{d}\tau} h(\tau)^{-1} \right) \,. \tag{A.17}$$

Note that $\widetilde{Z}(\tau = 0) = Z[n = 1]$, with $Z[n]$ being the original path integral over the $\psi$ variables in the main text. This is because when $n = 1$, the gauge field is zero, and so $Z[n = 1]$ and $\widetilde{Z}(\tau = 0)$ are both free fermion path integrals with the same boundary conditions.

So now our goal is to compute $\text{Tr}(\frac{dh}{d\tau}h^{-1})$, and then do the integral on the right hand side of (A.17). A major difficulty is that $h$ is an infinite matrix and its determinant or traces must be regularised. Our strategy is to find explicit expressions for the matrix elements of $h$, take the large $L$ limit and then construct the inverse and compute $\text{Tr}(\frac{dh}{d\tau}h^{-1})$. The final result is given in (A.46).

Computing the matrix elements of $h$ is straightforward. In the large $L$ limit, using (A.11), we get for instance,

$$h_{Am,An}(\tau) = \frac{\sqrt{2}}{T} \int\limits_{-T/2}^{T/2} dx_2 \; e^{i(w_m - w_n)x_2} \begin{cases} e^{\alpha(x_2)} & \text{if } w_n > 0 \\ -|c_1|^2 - |c_3|^2 e^{\alpha(x_2)} & \text{if } w_n < 0 \end{cases}, \tag{A.18}$$

where, for brevity and following [11], we defined

$$\alpha(x_2) := \tau H(x_2). \tag{A.19}$$

Other similar calculations give us the matrix

$$h(\tau) = \sqrt{2} \begin{pmatrix} [e^{\alpha}]_{++} & -|c_3|^2 [e^{\alpha}]_{+-} & 0 & c_1 c_3^* [e^{\alpha}]_{+-} \\ [e^{\alpha}]_{-+} & -|c_3|^2 [e^{\alpha}]_{--} - |c_1|^2 \, \mathbb{1} & 0 & c_1 c_3^* [e^{\alpha}]_{--} - c_1 c_3^* \, \mathbb{1} \\ 0 & c_1^* c_3 [e^{\alpha}]_{+-} & \mathbb{1} & -|c_1|^2 [e^{\alpha}]_{+-} \\ 0 & c_1^* c_3 [e^{\alpha}]_{--} - c_1^* c_3 \, \mathbb{1} & 0 & -|c_1|^2 [e^{\alpha}]_{--} - |c_3|^2 \, \mathbb{1} \end{pmatrix}, \tag{A.20}$$

where the row and column indices are valued in $\{A, B\} \times \{+, -\} = \{A+, A-, B+, B-\}$ and each entry in (A.20) is itself an infinite matrix with rows and columns indexed by the *positive* integers. Essentially, because the positive and negative frequencies behave differently, as in (A.18), we need to treat them separately. Also, following [11], we have introduced a square-bracket notation for the matrix elements in the Matsubara basis

$$[f]_{m,n} = \frac{1}{T} \int\limits_{-T/2}^{T/2} dx_2 \; e^{i(w_m - w_n)x_2} f(x_2). \tag{A.21}$$

It is important to note that this is a Toeplitz matrix since $[f]_{m,n} = [f]_{m-n}$.

The derivative $dh/d\tau$ and the inverse $h^{-1}$ can be obtained using the identities given in appendix B of [11]. After a brute force explicit calculation, we find that the $h^{-1}$ is composed of the block matrices

$$\sqrt{2} \, (h^{-1})_{A+,A+} = ([e^{\alpha}]_{++})^{-1} [M]_{++}, \tag{A.22}$$

$$\sqrt{2} \, (h^{-1})_{A+,A-} = -|c_3|^2 ([e^{\alpha}]_{++})^{-1} [M]_{++} [e^{\alpha}]_{+-} ([e^{\alpha}]_{--})^{-1}, \tag{A.23}$$

$$\sqrt{2} \, (h^{-1})_{A+,B+} = 0, \tag{A.24}$$

$$\sqrt{2} \, (h^{-1})_{A+,B-} = c_1 c_3^* ([e^{\alpha}]_{++})^{-1} [M]_{++} [e^{\alpha}]_{+-} ([e^{\alpha}]_{--})^{-1}, \tag{A.25}$$

$$\sqrt{2} \, (h^{-1})_{A-,A+} = \left( |c_1|^2 \mathbb{1} + |c_3|^2 ([e^{\alpha}]_{--})^{-1} \right) [N]_{--} [e^{\alpha}]_{-+} ([e^{\alpha}]_{++})^{-1}, \tag{A.26}$$

$$\sqrt{2} \, (h^{-1})_{A-,A-} = - \left( |c_1|^2 \mathbb{1} + |c_3|^2 ([e^{\alpha}]_{--})^{-1} \right) [N]_{--}, \tag{A.27}$$

$$\sqrt{2} \, (h^{-1})_{A-,B+} = 0, \tag{A.28}$$

$$\sqrt{2}\,(h^{-1})_{A-,B-} = \frac{c_1}{c_3}\left(|c_1|^2\mathbb{1} + |c_3|^2([e^\alpha]_{--})^{-1}\right)[N]_{--} - \frac{c_1}{c_3}\mathbb{1}\,, \tag{A.29}$$

$$\sqrt{2}\,(h^{-1})_{B+,A+} = -c_1^*c_3[e^\alpha]_{+-}([e^\alpha]_{--})^{-1}[N]_{--}[e^\alpha]_{-+}([e^\alpha]_{++})^{-1}\,, \tag{A.30}$$

$$\sqrt{2}\,(h^{-1})_{B+,A-} = c_1^*c_3[e^\alpha]_{+-}([e^\alpha]_{--})^{-1}[N]_{--}\,, \tag{A.31}$$

$$\sqrt{2}\,(h^{-1})_{B+,B+} = \mathbb{1}\,, \tag{A.32}$$

$$\sqrt{2}\,(h^{-1})_{B+,B-} = -|c_1|^2[e^\alpha]_{+-}([e^\alpha]_{--})^{-1}[N]_{--}\,, \tag{A.33}$$

$$\sqrt{2}\,(h^{-1})_{B-,A+} = c_1^*c_3\left(\mathbb{1} - ([e^\alpha]_{--})^{-1}\right)[N]_{--}[e^\alpha]_{-+}([e^\alpha]_{++})^{-1}\,, \tag{A.34}$$

$$\sqrt{2}\,(h^{-1})_{B-,A-} = -c_1^*c_3\left(\mathbb{1} - ([e^\alpha]_{--})^{-1}\right)[N]_{--}\,, \tag{A.35}$$

$$\sqrt{2}\,(h^{-1})_{B-,B+} = 0\,, \tag{A.36}$$

$$\sqrt{2}\,(h^{-1})_{B-,B-} = -\mathbb{1} + |c_1|^2\left(\mathbb{1} - ([e^\alpha]_{--})^{-1}\right)[N]_{--}\,, \tag{A.37}$$

where we have defined

$$[M]_{++} := \left(\mathbb{1} - |c_3|^2[e^\alpha]_{+-}([e^\alpha]_{--})^{-1}[e^\alpha]_{-+}([e^\alpha]_{++})^{-1}\right)^{-1}\,, \tag{A.38}$$

$$[N]_{--} := \left(\mathbb{1} - |c_3|^2[e^\alpha]_{-+}([e^\alpha]_{++})^{-1}[e^\alpha]_{+-}([e^\alpha]_{--})^{-1}\right)^{-1}\,. \tag{A.39}$$

Using $\frac{\mathrm{d}h}{\mathrm{d}\tau}$ and $h^{-1}$ so computed, we get

$$\tau\,\mathrm{Tr}\left(\frac{\mathrm{d}h}{\mathrm{d}\tau}h^{-1}\right) = |c_1|^2\,\mathrm{Tr}\left([\alpha]_{-+}[e^\alpha]_{+-}([e^\alpha]_{--})^{-1}[N]_{--}\right)$$

$$+ |c_1|^2\,\mathrm{Tr}\left([\alpha]_{+-}[e^\alpha]_{-+}([e^\alpha]_{++})^{-1}[M]_{++}\right)\,. \tag{A.40}$$

We can simplify this further by using the following identities for quantities that appear in the definitions (A.38) and (A.39)

$$[e^\alpha]_{+-}([e^\alpha]_{--})^{-1}[e^\alpha]_{-+}([e^\alpha]_{++})^{-1} = \mathbb{1} - \left([e^{-\alpha}]_{++}\right)^{-1}([e^\alpha]_{++})^{-1}\,, \tag{A.41}$$

$$[e^\alpha]_{-+}([e^\alpha]_{++})^{-1}[e^\alpha]_{+-}([e^\alpha]_{--})^{-1} = \mathbb{1} - \left([e^{-\alpha}]_{--}\right)^{-1}([e^\alpha]_{--})^{-1}\,, \tag{A.42}$$

and perform a binomial series expansion, to get[9]

$$\tau\,\mathrm{Tr}\left(\frac{\mathrm{d}h}{\mathrm{d}\tau}h^{-1}\right) = \sum_{p=0}^{\infty}(-1)^{p+1}|c_1/c_3|^{2p+2}\,\mathrm{Tr}\left([\alpha]_{-+}[e^\alpha]_{++}\left([e^{-\alpha}]_{++}[e^\alpha]_{++}\right)^p[e^{-\alpha}]_{+-}\right)$$

$$+ \sum_{p=0}^{\infty}(-1)^{p+1}|c_1/c_3|^{2p+2}\,\mathrm{Tr}\left([\alpha]_{+-}[e^\alpha]_{--}\left([e^{-\alpha}]_{--}[e^\alpha]_{--}\right)^p[e^{-\alpha}]_{-+}\right) \tag{A.43}$$

$$= \tau\frac{\mathrm{d}}{\mathrm{d}\tau}\,\mathrm{Tr}\log\left(\mathbb{1} + |c_1/c_3|^2[e^\alpha]_{--}[e^{-\alpha}]_{--}\right) \tag{A.44}$$

$$= \tau\frac{\mathrm{d}}{\mathrm{d}\tau}\,\mathrm{Tr}\log\left(\mathbb{1} - |c_1|^2[e^\alpha]_{-+}[e^{-\alpha}]_{+-}\right)\,. \tag{A.45}$$

---

[9] One needs to use identities of the form $[e^\alpha]_{--}[e^{-\alpha}]_{--} + [e^\alpha]_{-+}[e^{-\alpha}]_{+-} = \mathbb{1}$, see appendix B of [11] for more on such relations.

Recalling that $r = |c_1|^2$ is the reflection coefficient and $\alpha(x_2) = \tau H(x_2)$, we get

$$\text{Tr}\left(\frac{\mathrm{d}h}{\mathrm{d}\tau}h^{-1}\right) = -\frac{\mathrm{d}}{\mathrm{d}\tau}\sum_{m=1}^{\infty}\frac{r^m}{m}\,\text{Tr}\left(\left([e^{\tau H(x_2)}]_{-+}[e^{-\tau H(x_2)}]_{+-}\right)^m\right). \tag{A.46}$$

Substituting (A.46) into (A.17) and doing the $\tau$ integral, we get the result for the functional determinant

$$\log\left(\frac{\widetilde{Z}(1)}{\widetilde{Z}(0)}\right) = -\sum_{m=1}^{\infty}\frac{r^m}{m}\,\text{Tr}\left(\left([e^H]_{-+}[e^{-H}]_{+-}\right)^m\right). \tag{A.47}$$

Notice that the lower limit of the integral in (A.17) at $\tau = 0$ does not contribute on the right hand side, because the indices of the matrices $[e^\alpha]_{-+}$ and $[e^{-\alpha}]_{+-}$ are purely off-diagonal and so $[1]_{+-}$ and $[1]_{-+}$ vanish.

We were unable to evaluate the expression on the right hand side of (A.47) in general. Thus, we will focus on computing the functional determinant $\widetilde{Z}_k$ for the case needed in section 3, which is a single interval $[-L_-, L_+]$ across the defect. We will proceed in two different ways:

1. In appendix A.1 we only consider the linear in $r$ term in (A.47) and obtain an expression for a general interval $[-L_-, L_+]$. The result is given in (A.56).

2. In appendix A.2 we assume that $\gamma := \frac{L_- - L_+}{L_- + L_+}$ is small. This says that the interval is almost symmetric about the defect, and so we are perturbing away from the symmetrically placed interval considered in [8]. We obtain the Rényi and the von Neumann entropies as a power series expansion in $\gamma$, the results are given in (A.67) and (A.68).

## A.1 Perturbing away from the fully transmitting case

The goal of this section is to obtain the first order term in $r$ in the von Neumann entropy of the interval $[-L_-, L_+]$, perturbing away from the fully transmitting case $r = 0$.

The coefficient of the linear in $r$ term in (A.47) is

$$\text{Tr}\left([e^H]_{-+}[e^{-H}]_{+-}\right) = \sum_{n\geqslant 0, l<0}\int\frac{\mathrm{d}y_1\mathrm{d}y_2}{T^2}e^{\frac{2\pi\mathrm{i}}{T}((y_1-y_2)(l-n))}e^{H(y_1)-H(y_2)} \tag{A.48}$$

$$= -\int_{\mathbb{R}^2}\frac{\mathrm{d}y_1\mathrm{d}y_2}{4\pi^2}\frac{e^{H(y_1)-H(y_2)}}{(y_1-y_2)^2}, \tag{A.49}$$

where we have taken the $T \to \infty$ limit and the contours of integration for $y_1$ and $y_2$ are such that $\Im y_1 < 0$ and $\Im y_2 > 0$, so the geometric series in (A.48) converges. If we now expand in powers of $H$, we see that only terms at second order or higher will contribute, because for the constant and first order term, there is always one integral that vanishes after we close the contour in the respective half planes (the $y_1$ integral is closed in the lower half plane and the $y_2$ integral in the upper half plane).

Let us denote the $O(r)$ term in (A.47) by $f_k^{(1)}$ (with $k$ being the replica index), then

$$f_k^{(1)} = r \int_{\mathbb{R}^2} \frac{\mathrm{d}y_1 \mathrm{d}y_2}{4\pi^2 (y_1 - y_2)^2} \left( e^{H_k(y_1) - H_k(y_2)} - 1 - H_k(y_1) + H_k(y_2) \right). \tag{A.50}$$

To compute the entropy, we plug in the expression for $H_k$ from (2.35) and (2.23), which we write as $H_k = -2\Phi_k = \frac{k}{n} \log \phi$, with

$$\phi(y) = \frac{y^2 + L_-^2}{y^2 + L_+^2}. \tag{A.51}$$

We now sum over $k$ and take the $n = 1$ limit,

$$S^{(1)} = \lim_{n \to 1} \frac{1}{1-n} \sum_{k=-(n-1)/2}^{(n-1)/2} f_k^{(1)} = r \int_{\mathbb{R}^2} \frac{\mathrm{d}y_1 \mathrm{d}y_2}{4\pi^2 (y_1 - y_2)^2} \left( 1 - \frac{1}{2} \frac{\phi(y_1) + \phi(y_2)}{\phi(y_1) - \phi(y_2)} \log \left( \frac{\phi(y_1)}{\phi(y_2)} \right) \right). \tag{A.52}$$

Because of the logarithm in (A.52), we get branch cuts in the complex $y_1$ and $y_2$ planes. Let us assume, without loss of generality, that $L_- > L_+$. Since $y_1$ integration contour has a negative imaginary part, we deform the contour in the lower half plane and pick up a discontinuity from the branch cut that runs from $y_1 = -iL_-$ to $y_2 = -iL_+$. The discontinuity of the integrand of (A.52) is

$$-\frac{i\pi}{L_-^2 - L_+^2} \frac{2y_1^2 y_2^2 + 2L_+^2 L_-^2 + (L_+^2 + L_-^2)(y_1^2 + y_2^2)}{4\pi^2 (y_1 - y_2)^2 (y_1^2 - y_2^2)}, \tag{A.53}$$

which when integrated over $y_1$ from $y_1 = -iL_-$ to $y_1 = -iL_+$ gives

$$-\frac{1}{16\pi (L_-^2 - L_+^2) y_2^3} \left( L_-^2 L_+^2 \pi^2 - 2L_- L_+ (L_- - L_+) y_2 + (L_-^2 + L_+^2) \pi y_2^2 - 2(L_- - L_+) y_2^3 + \pi y_2^4 \right.$$

$$\left. + i(L_-^2 + y_2^2)(L_+^2 + y_2^2) \log \left( \frac{(y_2 + iL_+)(iL_- - y_2)}{(y_2 - iL_+)(iL_- + y_2)} \right) \right). \tag{A.54}$$

Now, we need to do the $y_2$ integral, whose contour has a small positive imaginary part and can be deformed in the upper half-plane. The discontinuity comes from the logarithm in (A.54) and equals

$$-\frac{1}{8} \frac{(L_-^2 + y_2^2)(L_+^2 + y_2^2)}{(L_-^2 - L_+^2) y_2^3}. \tag{A.55}$$

Integrating this over $y_2$ from $y_2 = iL_+$ to $y_2 = iL_-$ gives

$$S^{(1)} = \frac{r}{8} \left( 1 + \frac{L_-^2 + L_+^2}{L_-^2 - L_+^2} \log \frac{L_+}{L_-} \right). \tag{A.56}$$

Thus, to $O(r)$, the entropy is given by

$$S([-L_-, L_+]) = \frac{1}{3} \log \frac{L_+ + L_-}{\varepsilon} + \frac{r}{8} \left( 1 + \frac{L_-^2 + L_+^2}{L_-^2 - L_+^2} \log \frac{L_+}{L_-} \right) + O(r^2). \tag{A.57}$$

Higher order terms in the reflection coefficient can be computed in a similar way, but involve more integrals and a more complicated branch cut structure.

## A.2 Perturbing in the asymmetry

In this appendix, we will perform a different approximation. While still working with a single interval $[-L_-, L_+]$ that straddles the defect, we will compute the functional integral $\tilde{Z}_k$ perturbatively in the asymmetry parameter

$$\gamma = \frac{L_- - L_+}{L_- + L_+} = \frac{L_- - L_+}{2L}, \tag{A.58}$$

where $2L = L_- + L_+$ is the length of the interval.

We start with the following series expansion for $H_k$,

$$H_k(y) = -2\Phi_k(0, y) = \frac{2k}{n} \sum_{j \in \text{odd}} \frac{(\mathrm{i}\gamma L)^j}{j} \left( \frac{1}{(y + \mathrm{i}L)^j} - \frac{1}{(y - \mathrm{i}L)^j} \right). \tag{A.59}$$

The advantage of this expansion is that at any finite order in $\gamma$, we have poles in the complex plane instead of branch cuts.

The $m = 1$ term of (A.47) is given by

$$\mathrm{Tr}\left([e^{H_k}]_{-+}[e^{-H_k}]_{+-}\right) = \frac{1}{T^2} \int\limits_{-T/2}^{T/2} \mathrm{d}y_1 \mathrm{d}y_2 \, \frac{e^{H_k(y_1) - H_k(y_2)}}{\sin^2(\pi(y_2 - y_1)/T)}, \tag{A.60}$$

where $\Im y_1 < 0$ and $\Im y_2 > 0$ as mentioned earlier.

To evaluate these integrals, we will use the residue theorem. We close the contour for $y_1$ in the lower half-plane and for $y_2$ in the upper half-plane. This ensures that the only residues come from the pole at $y_1 = -\mathrm{i}L$ for the $y_1$ integral and the pole at $y_2 = \mathrm{i}L$ for the $y_2$ integral. We also get some contribution from integrals that were used to close the contours in the complex plane but these contributions vanish in the $T \to \infty$ limit.

In the limit $T \to \infty$, we have the leading order result in $\gamma$

$$\mathrm{Tr}\left([e^{H_k}]_{-+}[e^{-H_k}]_{+-}\right) = \frac{k^2}{n^2}\gamma^2\tau^2 + O(\gamma^4). \tag{A.61}$$

It is worth mentioning that the final answer must be even in $\gamma$. This is indeed the case for the above calculation.

In general, we want to compute $\mathrm{Tr}\left(([e^{H_k}]_{-+}[e^{-H_k}]_{+-})^p\right)$. In the $T \to \infty$ limit, this is given by a contour integral

$$\mathrm{Tr}\left(([e^{H_k}]_{-+}[e^{-H_k}]_{+-})^p\right) = \frac{(-1)^p}{\pi^{2p}} \int\limits_{-\infty}^{\infty} \prod_{i=1}^{2p} \mathrm{d}y_i \, \frac{e^{H_k(y_1) - H_k(y_2) + \cdots + H_k(y_{2p-1}) - H_k(y_{2p})}}{(y_2 - y_1) \ldots (y_{2p} - y_{2p-1})(y_1 - y_{2p})}, \tag{A.62}$$

where $\Im y_1, \Im y_3, \ldots \Im y_{2p-1} < 0$ and $\Im y_2, \Im y_4, \ldots \Im y_{2p} > 0$. We close the contours for $y_1, y_3, \ldots y_{2p-1}$ in the lower half plane and the contours for $y_2, y_4, \ldots y_{2p}$ in the upper half plane. This ensures that the contour does not enclose the poles coming from the $y_{i+1} - y_i$ terms in the denominator.

For $H_k$ given in (A.59), each factor of $\gamma$ corresponds to exactly one pole in the complex plane. Therefore, at order $\gamma^j$

$$\#(y_1) + \#(y_2) + \ldots \#(y_{2p}) = j\,, \tag{A.63}$$

where $\#(y_i)$ denotes the total number of poles (with multiplicity) for the $y_i$ variable in the numerator $e^{H_k(y_1) - H_k(y_2) + \cdots + H_k(y_{2p-1}) - H_k(y_{2p})}$. If $j < 2p$, there is at least one variable that has no pole in the entire complex plane. If we do the contour integral over this variable first, the result is zero. Therefore, $\mathrm{Tr}\left(\left([e^{H_k}]_{-+}[e^{-H_k}]_{+-}\right)^p\right)$ only contributes non-trivially starting at $O(\gamma^{2p})$.

To summarize, the $\gamma^j$ term in $\log\left(\widetilde{Z}_k/Z[1]\right)$ vanishes if $j$ is odd. If $j$ is even, we get a non-zero contributions to the $\gamma^j$ term from $\mathrm{Tr}\left(\left([e^{H_k}]_{-+}[e^{-H_k}]_{+-}\right)^p\right)$ only if $p = 1, 2, \ldots j/2$. Using (A.61), we have the full result at order $\gamma^2$,

$$\log \frac{\widetilde{Z}_k}{Z[1]} = -\gamma^2 \frac{k^2}{n^2} r + O\left(\gamma^4\right)\,, \tag{A.64}$$

so the Rényi entropy is

$$S_n\left([-L_-, L_+]\right) = \frac{n+1}{6n}\left[\log \frac{L_+ + L_-}{\varepsilon} - \frac{\gamma^2 r}{2}\right] + O\left(\gamma^4\right)\,. \tag{A.65}$$

Similarly, we can determine the contribution at higher orders in $\gamma$,

$$\begin{aligned}
\log \frac{\widetilde{Z}_k}{Z[1]} =& r\frac{k^2}{n^2}\left(\gamma^2 + \frac{\gamma^4}{2} + \frac{\gamma^6}{3} + \frac{\gamma^8}{4}\right) + rt\frac{k^4}{n^4}\left(\frac{\gamma^4}{2} + \frac{\gamma^6}{2} + \frac{65\gamma^8}{144}\right) \\
&- rt(r-t)\frac{k^6}{n^6}\left(\frac{\gamma^6}{6} + \frac{\gamma^8}{4}\right) + rt\frac{k^6}{n^6}\frac{\gamma^8}{72} + rt(r-5t)(5r-t)\frac{k^8}{n^8}\frac{\gamma^8}{144} + O\left(\gamma^{10}\right)\,,
\end{aligned} \tag{A.66}$$

We can now easily do the sum over $k$ to get the Rényi entropies and also take the $n \to 1$ limit to get the entanglement entropy.

$$\begin{aligned}
S_n\left([-L_-, L_+]\right) =& \frac{n+1}{6n}\left[\log \frac{L_+ + L_-}{\varepsilon} + \frac{r}{2}\left(-\gamma^2 - \frac{\gamma^4}{2} - \frac{\gamma^6}{3} - \frac{\gamma^8}{4}\right)\right] \\
&+ \frac{(n+1)(7 - 3n^2)}{240n^3}\left[rt\left(\frac{\gamma^4}{2} + \frac{\gamma^6}{2} + \frac{65\gamma^8}{144}\right)\right] \\
&+ \frac{(n+1)(31 - 18n^2 + 3n^4)}{1344n^5}\left[rt(r-t)\left(\frac{\gamma^6}{6} + \frac{\gamma^8}{4}\right) - rt\frac{\gamma^8}{72}\right] \\
&+ \frac{(n+1)(381 - 239n^2 + 55n^4 - 5n^6)}{11520n^7}\left[rt(r-5t)(5r-t)\frac{\gamma^8}{144}\right] + O\left(\gamma^{10}\right)\,,
\end{aligned} \tag{A.67}$$

$$\begin{aligned}
S\left([-L_-, L_+]\right) =& \frac{1}{3}\log \frac{L_+ + L_-}{\varepsilon} - \frac{r}{6}\left(\gamma^2 + \frac{\gamma^4}{2} + \frac{\gamma^6}{3} + \frac{\gamma^8}{4}\right) + \frac{rt}{30}\left(\frac{\gamma^4}{2} + \frac{\gamma^6}{2} + \frac{65\gamma^8}{144}\right) \\
&+ \frac{rt}{42}\left[(r-t)\left(\frac{\gamma^6}{6} + \frac{\gamma^8}{4}\right) - \frac{\gamma^8}{72}\right] + \frac{rt(r-5t)(5r-t)}{30}\left(\frac{\gamma^8}{144}\right) + O\left(\gamma^{10}\right)\,.
\end{aligned} \tag{A.68}$$

It is worth mentioning that the series multiplying $r$ adds up to $-\frac{1}{6}\log(1 - \gamma^2) = -\frac{1}{6}\log \frac{4L_- L_+}{(L_+ + L_-)^2}$, so these terms linearly interpolate between the purely transmitting limit, $r = 0$, and the purely reflecting limit, $r = 1$. All the other terms vanish at these two limits, and represent the deviation from this linear interpolation.

# B  Purely reflecting case and perturbations away from it

In this appendix we provide a direct evaluation of the functional integral and Rényi entropy for the purely reflecting boundary condition. We also discuss perturbations away from this limit.

In section 2, we ended up with the functional integral $Z_k$ in (2.21) after doing the replica trick and decoupling the replicas by introducing a gauge field. Then, we decoupled the fermion from the background gauge field by performing a combination of gauge and chiral transformations in (2.24). In section 2.2, we performed a single transformation on the entire plane $\Omega = \Omega^+ \cup \Omega^-$. This enabled us to compute the result in the purely transmitting case directly.

Instead of performing the gauge and chiral transformations given by (2.23), we now perform different transformations on the two half-planes. For later convenience, we take the $\Phi$-functions for the two half-planes to satisfy $\Phi_k^\pm(0, y_2) = 0$. It is easy to obtain such a $\Phi_k$ using the method of images

$$\Phi_k^+(\mathbf{x}) = -\frac{k}{n} \sum_{i=1}^{p^+} \log\left(\frac{|\mathbf{x} - \mathbf{u}_i^+||\mathbf{x} - \overline{\mathbf{v}}_i^+|}{|\mathbf{x} - \mathbf{v}_i^+||\mathbf{x} - \overline{\mathbf{u}}_i^+|}\right), \tag{B.1}$$

with a similar expression for $\Phi_k^-(\mathbf{x})$. Here, $\mathcal{A}^+ = \bigcup_{i=1}^{p^+}[u_i^+, v_i^+]$ refers to the subset of $\mathcal{A}$ on the positive real axis and $\mathcal{A}^- = \bigcup_{i=1}^{p^-}[u_i^-, v_i^-]$ refers to the subset of $\mathcal{A}$ on the negative real axis. The $\eta$-functions can be obtained using (2.18) and (2.22),

$$\eta_k^+(\mathbf{x}) = -\frac{k}{n}\left[\sum_{i=1}^{p^+}\left(\tan^{-1}\frac{x_2}{x_1 + u_i^+} - \tan^{-1}\frac{x_2}{x_1 + v_i^+}\right) + \sum_{i=1}^{p^-}\left(\tan^{-1}\frac{x_2}{x_1 - u_i^-} - \tan^{-1}\frac{x_2}{x_1 - v_i^-}\right)\right], \tag{B.2}$$

with a similar expression for $\eta_k^-(\mathbf{x})$.

The Jacobian for the transformation (2.24) with this choice of $\Phi_k$ and $\eta_k$ is

$$\log J_k^+ = -\frac{2k^2}{n^2}\,\xi\left(\{u_i^+\}, \{v_j^+\}\right), \quad \log J_k^- = -\frac{2k^2}{n^2}\,\xi\left(\{u_i^-\}, \{v_j^-\}\right), \tag{B.3}$$

where we have defined

$$\xi\left(\{u_i\}, \{v_j\}\right) = \Xi\left(\{u_i\}, \{v_j\}\right) - \frac{1}{2}\sum_{i,j}\log\left(\frac{|u_i + v_j||v_i + u_j|}{|u_i + u_j||v_i + v_j|}\right). \tag{B.4}$$

The quantity $\Xi$ was defined in (2.31) [5]. The functional integral thus becomes $Z_k = J_k^+ J_k^- \tilde{\tilde{Z}}_k$, where

$$\tilde{\tilde{Z}}_k = \int D\overline{\zeta}_k^+ D\zeta_k^+ D\overline{\zeta}_k^- D\zeta_k^- \exp\left(-\int_{\Omega^+} d^2x\,\overline{\zeta}_k^+ \,\displaystyle{\not{\partial}}\zeta_k^+ - \int_{\Omega^-} d^2x\,\overline{\zeta}_k^- \,\displaystyle{\not{\partial}}\zeta_k^-\right). \tag{B.5}$$

Since $\Phi_k^\pm(0, x_2) = 0$, the boundary condition for the transformed fermionic field $\zeta_k = \exp(-i\eta_k - \gamma_*\Phi_k)\psi_k$ is

$$\mathcal{B}\begin{pmatrix} e^{i\tilde{H}_k(x_2)} & 0 & 0 & 0 \\ 0 & e^{i\tilde{H}_k(x_2)} & 0 & 0 \\ 0 & 0 & 1 & 0 \\ 0 & 0 & 0 & 1 \end{pmatrix}\zeta_k(0, x_2) = 0, \quad \text{with } \tilde{H}_k(x_2) := \eta_k^+(0, x_2) - \eta_k^-(0, x_2). \tag{B.6}$$

In the purely reflecting case corresponding to $\mathcal{S}^{\mathrm{r}}$ in (2.7), the boundary condition in (B.6) is equivalent to $\mathcal{B}\zeta_k(0, x_2) = 0$, and so so $\tilde{\tilde{Z}}_k = Z[1]$. Thus, the Rényi entropy is

$$S^{\mathrm{r}}(\mathcal{A}) = \frac{1}{3}\xi\left(\{u_i^+\}, \{v_j^+\}\right) + \frac{1}{3}\xi\left(\{u_i^-\}, \{v_j^-\}\right). \tag{B.7}$$

This expression agrees with the results in [7] and, as expected, breaks up into a sum of two terms that just depend on $\mathcal{A}_+$ and $\mathcal{A}_-$ respectively.

With a nonzero transmission coefficient, a calculation similar to that in appendix A gives us the result analogous to (A.47)

$$\log\frac{\tilde{\tilde{Z}}_k}{Z[1]} = -\sum_{p=1}^{\infty}\frac{t^p}{p}\operatorname{Tr}\left(\left([e^{\mathrm{i}\tilde{H}_k}]_{-+}[e^{-\mathrm{i}\tilde{H}_k}]_{+-}\right)^p\right). \tag{B.8}$$

Again, we restrict ourselves to computing the functional determinant $\tilde{\tilde{Z}}_k$ which corresponds to a single interval across the defect, $\mathcal{A} = [-L_-, L_+]$.

Similar to appendix A.1, can compute the $O(t)$ piece in the von Neumann entropy by perturbing away from the fully reflecting case. The result is

$$S^{(1)} = t\int_{\mathbb{R}^2}\frac{dy_1 dy_2}{4\pi^2(y_1 - y_2)^2}\left(1 - \frac{1}{2}\frac{\phi(y_1) + \phi(y_2)}{\phi(y_1) - \phi(y_2)}\log\left(\frac{\phi(y_1)}{\phi(y_2)}\right)\right). \tag{B.9}$$

where

$$\phi(x) = \frac{(x - \mathrm{i}L_-)(x + \mathrm{i}L_+)}{(x + \mathrm{i}L_-)(x - \mathrm{i}L_+)}. \tag{B.10}$$

Using a similar contour deformation argument as in A.1, we find,

$$S^{(1)} = \frac{t}{8}\left(1 + \frac{L_-^2 - 6L_- L_+ + L_+^2}{2(L_- - L_+)\sqrt{L_- L_+}}\arctan\left(\frac{L_- - L_+}{2\sqrt{L_- L_+}}\right)\right). \tag{B.11}$$

The full answer for the entropy to order $t^2$ is then

$$S([-L_-, L_+]) = \frac{1}{6}\log\frac{4L_+ L_-}{\varepsilon^2} + \frac{t}{8}\left(1 + \frac{L_-^2 - 6L_- L_+ + L_+^2}{2(L_- - L_+)\sqrt{L_- L_+}}\arctan\left(\frac{L_- - L_+}{2\sqrt{L_- L_+}}\right)\right) + O(t^2). \tag{B.12}$$

We can also do a small $\gamma$ expansion similar to that in appendix A.2. We just quote the final result

$$\begin{aligned}
S([-L_-, L_+]) =& \frac{1}{6}\log\frac{4L_+ L_-}{\varepsilon^2} + \frac{t}{6}\left(\gamma^2 + \frac{\gamma^4}{2} + \frac{\gamma^6}{3} + \frac{\gamma^8}{4}\right) + \frac{rt}{30}\left(\frac{\gamma^4}{2} + \frac{\gamma^6}{2} + \frac{65\gamma^8}{144}\right) \\
&+ \frac{rt}{42}\left[(r - t)\left(\frac{\gamma^6}{6} + \frac{\gamma^8}{4}\right) - \frac{\gamma^8}{72}\right] + \frac{rt(r - 5t)(5r - t)}{30}\left(\frac{\gamma^8}{144}\right) + O\left(\gamma^{10}\right).
\end{aligned} \tag{B.13}$$

This matches with (A.68) at the appropriate order in $\gamma$.

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
