# Peer review of "Free fermion entanglement with a semitransparent interface: the effect of graybody factors on entanglement islands"

_SciPost Physics_

## Round 1 · Referee Report · Anonymous · 2021-7-28

Strengths

1.) A simple toy model that illuminates difficult to calculate effects in an important problem
2.) Connects established results from many body statistical mechanics to a modern topic in quantum gravity

Weaknesses

1.) Discusses graybody factors of black hole evaporation, but limited in scope to perturbative results at zero temperature
2.) Ignores subtleties about boundary conditions at the defect which may be important

Report

The manuscript presents two novel results: first, the entanglement entropy of asymmetric intervals about a defect in a theory of relativistic massless two-dimensional fermions and second, the location of the quantum extremal surface in a two-dimensional theory of quantum gravity coupled to a bath via semi-transparent boundary conditions.

The first result is an extension of work performed in the statistical mechanics community where results were confined to either manifolds with a boundary or symmetric intervals surrounding the defect. In order to derive the new results, the authors use the replica trick and evaluate a Euclidean path integral with n copies of the original fields that are subject to boundary conditions inherited from the n-sheeted replica path integral. These boundary conditions can be absorbed into a (non-unique) background gauge field which can then be removed by combined gauge and chiral transformation of the fermions which leads to novel boundary conditions for the transformed fermions. While standard in the statistical mechanics literature, the asymmetry of the entangling region and the non-vanishing of both the reflection and transmission coefficients provide a technical challenge in implementing these boundary conditions which is well-described in the appendix of the manuscript. Impressively, for sufficiently small values of the reflection or transmission coefficients, or of the asymmetry of the interval, the authors are able to derive perturbative formulae for the Renyi and entanglement entropies.

My few reservations with this section of the paper are the following. First, as pointed out in Refs. [7] and [8], the defect allows for two different boundary conditions that break either an axial or a vector U(1) symmetry of the theory, respectively. I would like to see a discussion in the manuscript regarding the impact of these boundary conditions on the various fields used in the replica calculation. While it is not obvious from Refs. [7] and [8] that they play a role in the symmetric interval entanglement entropy, they do play a role in the modular Hamiltonian, and it is not obvious that asymmetry of the interval does not cause the two phases to differ in their entanglement entropies. Second, a minor point is that the smallness of the various parameters should be stated in reference to some other dimensionless quantity. This potentially is important for the gravitational story, as I discuss further below.

With the entropy result in hand, the authors turn to the problem of finding quantum extremal surfaces in a theory of zero temperature JT gravity coupled to a two-dimensional bath of massless fermions in their ground state without gravity. Considering the defect to be a 1-dimensional quantum system living at the interface between the gravitational system and the bath, this requires the entanglement entropy calculation from the first section. This section is a very nice application of the technical achievements of the first section.

As in the earlier section, the smallness of certain parameters should be made more precise. For instance, in eq. (3.9), in the limit of large b, if t is parametrically larger than k/b, though still small, then a is smaller than b. But from eq. (7) of Ref. [20], a~b for k/b much less than one. It seems, then, that there could be a range of small t where the QES is actually closer to the boundary than when t = 1. It would be nice to see a more thorough discussion of this point, especially in consideration of the conclusion that graybody effects are monotonic. A few other small points: much of this discussion relies heavily on the fact that the fermions are in the ground state and the gravitational system is at zero temperature. A short sentence discussing why the results of the first section are immediately applicable without modification to the gravitational calculation could be useful, since the fermions live in AdS_2 in the bulk. Finally, my understanding of the results of this section is that increasing graybody effects (meaning increasing r) moves the QES deeper into the bulk. This seems to contradict the final sentence since I would expect in the evaporating black hole case, the QES should move further inside the horizon due to these effects. If the authors could clarify this point in their response, that would be helpful.

Requested changes

1.) Discuss boundary conditions
2.) Discuss range of validity of perturbative expansion

---

## Round 1 · Referee Report · Anonymous · 2021-8-5

Strengths

1- Novel formulae are obtained for the von Neumann and Renyi entropies of 2d free fermions on an interval in the presence of an interface in new regimes of transparency and symmetry for which explicit results were not previously known.
2- Many interesting recent results on black hole evaporation and the information paradox ignore graybody factors - this paper presents a first attempt at modelling these effects in modern approaches to entanglement in quantum gravity.

Weaknesses

1- Discussion of certain aspects in section 3 is a bit superficial (e.g. no comments on flat-flat vs. AdS-flat interfaces or on temperatures T>0)
2- Poor motivation of possible future directions or extensions of this work

Report

This paper carries out a detailed calculation of entropic quantities in a 2-dimensional theory of free fermions. In particular, they consider an interval with an interface in between in various limits: an almost fully reflective / transmissive interface and an almost symmetric interval across the interface. Remarkably, they are able to obtain explicit perturbative results in these situations via an application of the replica trick with the introduction of background gauge fields, extending the treatment of [5] to their settings. This new technical result is then used in a toy model for black hole entanglement to explore the effects of graybody factors on quantum extremal surfaces (QES) in holography. The fermion interface plays the role of coupling gravity to a bath of free fermions across an AdS-flat boundary. In a zero-temperature model, they observe a monotonic recession of the QES towards the Poincare horizon as the reflection coefficient of the interface is increased, consistent with the intuition that the entanglement wedge of radiation should shrink as the bath decouples. On physical grounds, the authors conjecture that this monotonic behavior may be generic.

At the technical level, the results of the paper are interesting and very neat. There is just one observation of their results which may be worth a comment: equations (2.41) and (2.42) seem to exhibit divergences in the perturbative terms as L_- or L_+ is taken to be small (i.e., when the interval crosses the interface only slightly). Is this meaningful/expected, or does it hint at a breakdown of the perturbative approach to small r or t?

The application of their results to a calculation in quantum gravity is also noteworthy. However, despite their strong and careful technical results in section 2 and the appendices, section 3 lacks a more elaborate discussion of various aspects which seem important. For instance, is there any subtlety in applying the flat-flat interface results to the AdS-flat interface in the gravity setup? Why or why not? Also, the authors only consider JT at zero temperature - what is the motivation behind this or the difficulties one would face otherwise? Would the differences be qualitative or just quantitative? The paper would also benefit from some more discussion of possible future directions, particularly regarding the program of including graybody factors in black hole evaporation and their effect on quantum extremal surfaces (e.g. expectations at finite temperature / in higher dimensions?)

Requested changes

1- Comment on divergence of (2.41) and (2.42) as L_- or L_+ go to zero.
2- Comment on applicability of flat-flat results to AdS-flat interface.
3- Comment on subtleties and expectations for T>0

---

## Editorial Decision

resubmitted